# Evaluation of Nutritional and Health Status in Captive Eastern Indigo Snakes (*Drymarchon couperi*) in Response to Formulated Sausage Diet

**DOI:** 10.3390/ani14223324

**Published:** 2024-11-19

**Authors:** Peyton R. Jackson, James E. Bogan, Ellen S. Dierenfeld, Zachary J. Loughman

**Affiliations:** 1Department of Organismal Biology, Ecology, & Zoo Science, West Liberty University, West Liberty, WV 26074, USA; zloughman@westliberty.edu; 2Central Florida Zoo & Botanical Gardens’ Orianne Center for Indigo Conservation, Eustis, FL 32736, USA; jamesb@centralfloridazoo.org; 3Zootrition Consulting, St. Louis, MO 63128, USA; edierenfeld@aol.com; 4World Wildlife Fund, Washington, DC 20008, USA

**Keywords:** clinical pathology, diet formulation, *Drymarchon couperi*, evidence-based husbandry, herpetoculture, nutrition

## Abstract

Captive breeding and headstarting programs are useful conservation tools to assist imperiled species in recovery. The eastern indigo snake (EIS) benefits from managed breeding colonies, but there are concerns regarding their reproductive fitness given the incidence of complications during egg-laying in first-time mothers. Potential causes include body overconditioning or nutrient imbalances, yet little information is available regarding snake nutrition, as domestic rodents are regarded as a nutritionally complete diet irrespective of the snake’s natural history. In this study, we examined the health and nutritional status of adult EIS maintained on standard mixed-whole-prey diets or a “faux-snake” sausage diet aligned with the nutritional profile of the EIS’s preferred prey. While domestic rodents are significantly higher in fat and lower in vitamin E compared to snakes predated by free-ranging EIS, our study demonstrated that current mixed-prey diets appear sufficient to meet the vitamin E and selenium needs of EIS, although further investigation into the vitamin D_3_ status of captive snakes is warranted. There were no ill effects associated with the consumption of an atypical diet in an atypical format, and longitudinal studies with juvenile snakes are suggested to examine the influence of nutrient composition on reproductive health in this species.

## 1. Introduction

The eastern indigo snake (*Drymarchon couperi*, EIS) was first listed as Threatened under the Endangered Species Act in 1978, with ongoing listing approved in 2008 due to concerns regarding habitat degradation and fragmentation [1]. State and federal agencies, public institutions, and non-profit organizations are invested in conservation initiatives focused on protecting EIS habitat and reestablishing populations within its historic range through captive breeding and headstarting programs [2,3]. Although 269 snakes were released at two sites between 2010 and 2022, the output of the breeding colony was insufficient to meet the repatriation goal of releasing 30 two-year-old snakes per release site annually over a decade to minimize the risk of extinction at the reintroduction sites [3,4]. By examining the health and condition of managed breeding colonies, we can better understand the impact of husbandry practices to improve welfare and reach management milestones.

Current husbandry and management practices may predispose snakes to obesity and other conditions associated with high-fat diets and limited physical activity [5]. In herpetoculture, snakes are frequently maintained on ‘mono-diets’ of domestic rodents, as it is presumed that whole-prey items are sufficient to meet their dietary requirements [6,7]. Feeding protocols for herbivorous and insectivorous reptiles stress the importance of diet diversity and supplement schedules so that nutrients may accumulate over time [6]. Conversely, a mono-diet of rodents at a consistent and frequent schedule may allow nutrients like lipids to accumulate faster than the snake is capable of metabolizing them.

Obesity can also exacerbate nutrient imbalances by interfering with the metabolic processes that organisms rely on to maintain sufficient levels of fat-soluble compounds like cholecalciferol (25-hydroxy vitamin D_3_) and alpha-tocopherol (vitamin E). Both vitamins play crucial physiological roles by regulating calcium absorption in the intestines [8] or behaving as biological antioxidants to protect tissues [9]. Both vitamins have demonstrated impacts on the reproductive fitness of multiple species and occur at different levels in whole prey, influenced by species, diet, seasonality, and husbandry practices [9,10,11,12,13,14,15]. Vitamin D_3_ deficiencies can weaken the durability of eggshells and manifest as bone deformities, neurological disorders, and dysfunction of the muscles but can also remain undetected in affected individuals [16]. While bone deformities are typically irreversible, appropriate husbandry, like full-spectrum lighting, thermal gradients, and nutrient supplements, can resolve other complications [6,8,17,18].

Similar to aquatic turtles and other piscivores, snakes fed primarily fish-based diets are at a higher risk of nutrient imbalances due to inappropriate prey items or improper storage [17]. Frequently encountered prey items such as goldfish (*Carassius auratus*), shiners (*Notropis* sp.), and minnows (*Pimephales* sp.) contain enzymes that render thiamin (vitamin B_1_) metabolically inactive [19]. Unlike other deficiencies, this disorder has a rapid onset with clear clinical signs, including neurological dysfunction, whereas hypovitaminosis E may have a delayed onset of signs due to the compound’s fat-soluble nature [17,18]. Even with a nutritionally complete diet, inappropriate storage and chemical interactions between nutrients like vitamin E and rancid fish oils can degrade the nutritional quality of prey items. For example, a long-term frozen-stored diet of lesser sand eels (*Ammodytes tobianus*) was associated with the occurrence of coelomic steatitis in a group of tentacled snakes (*Erpeton tentaculatum*) over a six-year period. In addition to fat necrosis and inflammation in the lower third of the body, hepatic lipidosis was also identified in four of the six snakes [20]. Outside of specialized species that refuse standard mixed prey diets, managed snakes are typically not provided with supplements or access to UV-B under the assumption that domestic whole-prey items are a nutritionally complete meal, and snakes rarely present with clinical signs associated with hypovitaminosis D_3_ or other nutritional deficiencies [6,7].

Extensive research into the ecology and physiology of EISs has led to the development of protocols with their natural history in mind [21]. In the longleaf pine forests of the southeastern United States, EISs are opportunistic and active foragers, preying on reptiles, amphibians, small mammals, and birds [22,23]. In an assessment of 185 prey records, snakes, anurans, and juvenile gopher tortoises (*Gopherus polyphemus*) constituted over 85% of food consumed, and 49% of observations documented ophiophagous behavior [24]. To reflect this diverse diet, EISs at the Central Florida Zoo & Botanical Gardens’ Orianne Center for Indigo Conservation (OCIC) are fed a rotating diet of adult rodents, juvenile birds, fish, and frog legs [21,25]. Additionally, snakes are maintained in accordance with the seasonal temperatures of central Florida to aid in conditioning the snakes for breeding in the late fall and early winter.

The primary concern with the reproductive fitness of the breeding colony is the occurrence of dystocia, colloquially referred to as “eggbinding”, where the female is unable to finish passing the egg through the oviduct [11]. Dystocias may result from large eggs, contortions in the oviduct, or the result of metabolic issues or nutritional imbalances, including dehydration and calcium deficiencies [11,26]. A retrospective study of the 104 breeding events observed at the OCIC found that nulliparous dams were 9.2 times more likely to experience complications during oviposition, irrespective of age and weight [26]. Additional research into the health and nutritional statuses of EIS under current management practices may aid in the ongoing development of evidence-based husbandry protocols to improve the welfare of managed EIS breeding colonies by reducing the occurrence of eggbinding in future generations.

The goal of this pilot study was to duplicate the nutritional profile of wild prey items consumed by EISs and evaluate the effects of a ‘faux snake’ sausage diet on plasma concentrations of vitamins E and D_3_ and selenium to provide support for future longitudinal studies on the linkages between nutrition and reproductive output in EISs.

## 2. Materials and Methods

This study was conducted with the approval of the West Liberty University Institutional Animal Care and Use Committee (IACUC #2021-03) and the Central Florida Zoo & Botanical Gardens’ Research Committee (Project #2016-01). Nine adult EISs (3 males and 6 females aged between 4 and 7 years) captive-bred at the OCIC (Eustis, FL, USA) were included in the study based on their health history and reliability of consuming control and experimental diets. Given the diverse demographics of the OCIC’s breeding colony, snakes were randomly assigned to one of three diet treatments: the basal whole-prey diet (negative control), the basal sausage diet (positive control), or the experimental sausage. Each diet treatment included one male and two female snakes.

Husbandry for the snakes in this study remained consistent with that of the breeding colony, with the exception of feed type for snakes in the basal and experimental sausage groups. Throughout the study, select EISs were rotated between indoor and outdoor enclosures at the OCIC. Indoor enclosures consisted of rack-style cages with a water dish and appropriate hides and substrates according to regulations described by the Association of Zoos and Aquariums Taxon Advisory Group [25]. Husbandry (enclosure maintenance, health checks, and feeding) was performed by OCIC staff. Snakes were fed to appetite 1–3 times a week depending on body condition and seasonal cycling goals. EISs were seasonally cycled as a means of conditioning the animals for reproduction according to environmental conditions in central Florida. Ambient temperatures were maintained through the facility’s central heating and cooling, and additional heat was not provided. Outdoor enclosures consisted of screen-enclosed pens inside a larger screened-in area. These pens provide the snake with access to filtered natural sunlight, sandy substrate, and naturalistic decor, such as plants, branches, and artificial burrows with keeper-access points (Figure 1). Snakes were moved to their indoor enclosures prior to sample collection and during inclement weather.

### 2.1. Diet Nutritional Analysis

The experimental sausage replicated the nutritional composition of the average profile of prey found in the stomach contents of free-ranging EISs as analyzed by Dierenfeld et al. (2015) [27]. Zootrition 2020 (St. Louis, MO, USA), a dietary management software with a database of published nutritional profiles and energetics of over 3000 feedstuffs for diet formulation, was used to develop a high-protein, low-fat sausage with commercial supplements. Samples were sent to a reference laboratory (Dairy-One Ithaca, NY, USA) to confirm the calorimetric and nutritional profile (dry matter, crude protein, crude fat, ash, macro- and trace minerals), while another reference laboratory (Midwest Labs Omaha, NE, USA) confirmed dietary profiles for vitamins A, D, and E; selenium; and amino acids. The recipe was revised twice following laboratory analyses (Table 1).

The final recipe consisted of low-fat white muscle meat (pork loin, rabbit tenderloin, and alligator tail filets) supplemented with dicalcium phosphorus, uniodized salt, and MeatComplete with Taurine supplement (Nebraska Brand, North Platte, NE, USA). The basal sausage did not receive additional supplements and duplicated the proportional composition of whole prey relative to their weight as fed out to the OCIC breeding colony.

### 2.2. Diet Preparation and Treatments

The methodology for sausage preparation was developed based on a protocol for making food-grade sausages. The meat was cut into cubes using a steel meat cleaver and ground with an STX International TurboForce 3000 electric meat grinder (Lincoln, NE, USA). Nutritional supplements for the experimental sausage were thoroughly incorporated by hand prior to the blend being stuffed into fresh beef-collagen casings. One end of the casing was tied off and stuffed with 50–60 g of meat. The casing was cut and tied off, with excess casing cut away above the knot (Figure 2). Casing dimensions of 22, 29, and 32 mm diameter were used throughout the study, with 29 mm being the ideal diameter based on the range of motion in the jaw of an average adult-size EIS. A more thorough methodology with images, tips, and recipes is available as Appendix A.

During the development of the sausage recipe and methodology, preliminary feeding trials were conducted with a gray rat snake (*Pantherophis spilotes*), yellow-tail cribo (*D. corais*), and false water cobra (*Hydrodynastes gigas*) that informed the final methodology for presenting sausage diets to snakes. During the first feeding trial, sausages were thawed directly in water. The gray rat snake struck and ruptured the casing, covering half the enclosure in ground meat. The keeper returned approximately 10 min later with cleaning supplies, but the rat snake had consumed the sausage filling. Other trials with the false water cobra and yellow-tail cribo showed that snakes known to have ravenous appetites readily accepted the sausage diet. When bagged and thawed in room-temperature water, the casings maintained their integrity throughout most feeding events. Twenty-three EISs at the OCIC were offered experimental sausages, and only three sausages ruptured. In two instances, minimal filling was lost to the environment. Snakes also showed interest in the filling outside of the sausage casing. In the third instance, the snake’s teeth sliced the casing, and in spite of approximately one-third of the contents falling out, the snake finished the meal.

### 2.3. Sample Collection and Analysis

Whole blood and plasma samples were collected from each snake every three months between August 2022 and September 2023 (n = 5 sampling events) by the attending veterinarian. Eastern indigo snakes were fasted three days prior to sample collection, and one of two methods was used in restraining snakes. In the baseline and fall collection events, snakes were sedated with 10 mg/kg alfaxalone IM (Alfaxan, Multidose IDX, Zoetis, Parsippany, NJ, USA) to facilitate phlebotomy. The use of sedatives was discontinued following the death of EIS #295 two days following the November collection visit. There was an advanced degree of autolysis present, which impeded the microscopic interpretation of tissues, especially of the viscera. The pathologist was unable to establish nutritional status or identify any lesions. Given the time of year, night-time lows were around 18 °C, and it is possible the snake was unable to fully metabolize the sedative due to its reduced metabolic activity. For the winter collection event and beyond, snakes were only manually restrained by technicians during the procedure.

A pre-heparinized 25-gauge, 1″ needle attached to a 3 mL syringe was used to draw blood from the snake’s ventral coccygeal vein while it was physically restrained by a technician. Blood was immediately placed into a 3 mL lithium heparin vacutainer tube (Greiner Bio-One, Fisher-Scientific, Monroe, NC, USA) and inverted several times for adequate mixing. One aliquot of whole blood was separated for hematology, and the remaining sample was centrifuged at 1327× *g* for 10 min.

Blood and plasma samples were submitted to three reference laboratories via overnight shipping with ice packs to maintain sample integrity. Whole blood samples and slides were kept separate from ice packs with insulated packing. Plasma samples for vitamin A/E analysis by high-performance liquid chromatography (HPLC) and vitamin D_2/3_ analysis by liquid chromatography–tandem mass spectrometry (LC/MS/MS) were submitted to Heartland Assays (Ames, IA, USA). Plasma samples were also submitted to Michigan State University Veterinary Diagnostic Laboratory (Lansing, MI, USA) for selenium analysis. Whole blood, plasma, and blood smears were sent to the University of Miami Avian & Wildlife Laboratory (Miami, FL, USA) for chemistry panels, complete blood counts, and protein electrophoresis.

### 2.4. Data Analysis

In addition to the complete blood count, over 22 analytes were examined in the plasma biochemistry analysis. Data collected during this study were examined in Microsoft Excel Version 2310 (Build 16923.20124) and summarized as means, medians, range, and standard error. The results of the vitamin D_3_ and E and selenium analysis are presented here, with the remaining data publicly available as Appendix A.

## 3. Results

Both basal and experimental sausages were well received by not only EISs but other colubrid and colubroid species as well. Snakes exhibited normal feeding behaviors, including pinning the sausage to the sides of the enclosure with their upper body and using their jaws to manipulate the prey into an optimal position for ingestion. In some cases, larger snakes were capable of folding sausages in half and consuming them from the center as opposed to one end. Instances in which snakes refused meals were easily corrected with the introduction of another prey item prior to the sausage diet. Keepers did not report any abnormalities in body condition, bowel movements, or behavior in snakes consuming either sausage diet.

Due to the small sample size of each dietary treatment, no formal traditional analyses were conducted. Instead, we only rely on descriptive statistics to interpret differences between study groups. We consider our results a pilot study and encourage readers to use and interpret them as such.

### Health and Nutritional Status

Given the extent of analytes examined in the hematology and plasma biochemistry assays, the results reported here only focus on the vitamin status of the nine EISs under the three dietary treatments and how the three treatments compare to the values observed in free-ranging EISs [28]. While no significant differences were observed between groups, the remaining results are still important for understanding the health of managed EISs and are publicly available as a Appendix A.

Plasma concentrations of vitamin E were well within and exceeded the free-range values (maximum = 0.0365 mg/mL; Figure 3), especially following the administration of supplemental vitamin E 1 month prior to the fall sampling event. Vitamin E was administered IM at [7.5 IU/kg] as another means of conditioning females in preparation for egg-laying. No other supplements are used in the OCIC’s breeding colony care protocols.

Selenium levels were also consistent among the three dietary treatments, with noticeably lower levels in winter and spring (see Figure 4). Although higher levels were observed in the summer collection event in both whole-prey groups, a similar rebound was not observed in snakes that were fed the experimental sausage diet.

The most obvious discrepancy between free-ranging and captive snakes was seen in plasma vitamin D_3_ concentrations (Figure 5). The only snakes to breach the minimum value of 46 ng/mL were EIS #409 (64.1 ng/mL) and #392 (60.9 ng/mL), which were housed outdoors prior to collection trips in November and January, respectively. Vitamin D_3_ levels dropped between the winter and spring sampling events for both whole-prey and basal sausage groups, while the mean summer values were lower than the baseline values recorded prior to the start of the dietary treatment for all snakes.

## 4. Discussion

This study examines the health and select nutritional statuses of a small group of federally threatened snakes with an established captive breeding and headstarting program to re-establish wild populations. Additionally, this paper compares the nutritional status of snakes (Figure 3, Figure 4 and Figure 5) fed mixed whole prey against a home-formulated diet (Table 2). While the full effects of the experimental diet and the potential link of nutrition to dystocias in EISs can only be fully realized through a multi-year study with initiated juvenile subjects, the physical condition of the animals and their health status suggests that there are no ill effects associated with the consumption of atypical prey in an atypical format and that the mechanical processing of meat does not affect nutrient composition or uptake, as both whole-prey and basal sausage groups had similar results.

A series of recommendations and guidelines for food items, supplements, and the frequency of meals offered are available regarding feeding husbandry for herbivorous and omnivorous reptiles [6]. However, snake nutritional protocols atypically suggest ‘appropriately sized prey’ fed at a more frequent rate for snakes with higher metabolisms, like juvenile snakes and colubrids, or at longer intervals (i.e., weekly) for those with lower metabolisms, such as geriatric snakes and boids [12,13]. Additionally, commercially produced formulated diets for amphibians, aquatic turtles, and other reptiles are available in gel, pellet, and sausage formats, although nutritional profiles or adequacy may not be fully described. Additional studies investigating alternative diets for snakes in recent years have developed low-calorie black soldier fly larvae sausages to address issues with rodent prey, like expense and calorie density [30]; other concerns in snake nutrition and feeding typically revolve around transitioning hatchlings or wild-caught snakes onto rodent-based diets [7,18,31].

Previous research has shown that there are differences in the nutritional content of free-range and domestic species frequently consumed by carnivores in human care, which could also contribute to overconditioning or nutrient imbalances in captive snakes (Table 2). An analysis of snakes found in the stomach contents of roadkilled EISs found that the average crude fat content was 7.28% of dry matter (DM) and the average crude protein content was 73.30% DM [28], while the crude fat and protein content of domestic rats are approximately 30% DM and 60% DM [10], respectively. In recent years, domestic chicks and day-old quail have been more frequently offered by vendors of whole prey for exotic wildlife (Table 2). Frog legs, typically sold at specialty grocery stores, are also becoming more popular to incorporate into the diets of amphibian- and fish-eating specialists. Although frog legs have a much higher ratio of protein to fat versus whole rodents (see Table 2), they are an inadequate primary source of essential nutrients like vitamins E and D_3_ and calcium due to the lack of organ tissue [29] and bone [32]. As prey items, snakes contain significantly higher concentrations of vitamin E than most domestic whole-prey options, especially avian prey. Nonetheless, the standard mixed whole-prey diet appears sufficient to meet the vitamin E and selenium requirements for EISs when using free-ranging values as a guideline (Figure 3).

While domestic whole-prey selections have lower levels of vitamin E in comparison to free-range prey items [10,27], rotating between multiple prey items with different profiles may prevent the development of nutritional deficiencies through the accumulation of nutrients over time [6]. With regard to the experimental sausage, snakes maintained higher levels of vitamin E following injections in October 2022 and throughout the remainder of the study. The maximum observed in this study (0.080 mg/mL) exceeded the maximum observed in free-ranging EISs (0.0365 mg/mL) [28] and values reported in the green anaconda (*Eunectes murinus*, 0.0124 mg/mL) [33] and eastern massasauga (*Sistrurus catenatus*, 0.0249 mg/mL) [34]. While selenium is an important co-factor for antioxidant function, the only other data available on selenium levels in snakes are values reported on *S. catenatus* (67.7 ng/kg, [35]; 107.45 ng/mL [34]), both of which are surpassed by the Se concentration measured in the groups of this study (Figure 4).

However, neither the whole-prey nor formulated diets were sufficient to meet the vitamin D_3_ needs of EISs when using free-ranging values as a guideline [28]. There is a large discrepancy between the values observed in this study and those reported by Knafo et al. (2016) [28], including instances in which snakes had access to UVB exposure in outdoor screen-enclosed pens. This is consistent with results published by Bogan et al. (2020) [36] regarding the effect of UVB exposure on the vitamin D_3_ status of managed EISs. This raises an important question regarding vitamin D_3_ uptake in snakes, as it is largely believed that they are capable of meeting their vitamin D_3_ requirements through whole-prey consumption alone [6,7].

Cholecalciferol (vitamin D_3_) is responsible for the regulation of calcium and phosphorus absorption in the intestine, and deficiencies typically manifest as bone deformities, neurological disorders, and muscular dysfunction in addition to poor reproductive health [6,8,17]. Though testudines and other squamates are provided with UVB lighting to aid in vitamin D_3_ synthesis in conjunction with a properly formulated diet that considers how vitamins and minerals interact, snakes are one of the few groups for which full-spectrum lighting or nutrient supplements are not encouraged nor widely practiced. Recent studies on corn snakes (*Pantherophis guttatus*) [37], Burmese pythons (*P. bivittatus*) [38], and eastern indigo snakes [36] have found that the provision of UVB increases circulating levels of vitamin D_3_ compared to unexposed groups, but contradicting work on ball pythons (*Python regius*) [39] failed to discern a significant difference. Further research is needed to standardize the provision of full-spectrum lighting so that we can better understand the vitamin D_3_ needs of snakes and their ability to sequester it from their diet, but this also requires an in-depth understanding of the focal species’ natural history and physiology given the importance of the animal’s natural condition to promote success in a captive environment.

While domestic rodents and birds, such as chicken and quail, are the primary whole prey available to captive carnivores, incorporating both into a snake’s diet may aid in avoiding overconditioning due to the lower fat content of chicks, while rodents can make up for the lower levels of vitamins A and E observed in avian whole prey [10]. Further recommendations for feeding snakes in captivity, such as the size and frequency of meals, should be informed by the animal’s natural metabolism, life stage, and health status while also considering the snake’s primary enclosure and opportunities for exercise outside of that space. Management decisions related to conservation, education, and research initiatives may also influence metabolic demands if the snake is an active breeder or nutrient intake if the caretakers initiate a training program with food rewards [40].

There are numerous extraneous variables that could influence the data collected over the course of this study. The subjects are active members of the OCIC’s breeding colony that produces EISs for reintroduction efforts and educational outreach with partner institutions, so their environmental conditions would change throughout the year, which, in turn, would affect their metabolism and appetite in addition to their hematology and plasma biochemistry [33,41]. Individual variations, such as demographics [42], past husbandry [43], ecdysis status [44], and illnesses or injuries [45,46,47], could further influence results and confound direct comparisons between dietary treatments. This stresses the need for longitudinal studies with larger groups so that greater control can be exercised over variables like husbandry and individual condition.

Sample collection was a limiting factor with respect to the sample size submitted for hematology and plasma biochemistry analysis. It is difficult to collect 3 mL of blood from EISs despite their large size. During phlebotomy, snakes constricted the flow of blood to the ventral coccygeal vein, and several attempts were needed to collect 1–2 mL. In some situations, multiple attempts were made to draw blood after a short period of rest or at different sites, such as the palatine vein in the roof of the snake’s mouth. Differences in collection site and timing add an additional layer of uncertainty when drawing conclusions from the results presented. Additionally, the elevated plasma total calcium concentrations of EISs hindered sample analysis due to the series of dilutions needed and realistic expectations regarding sample size [48]. This study still generated a valuable bank of information on EIS health and nutrition statuses that will help inform future sampling methods and research objectives.

## 5. Conclusions

With the conclusion of this pilot study, we have gathered a year of data on the health status of a small subset of EISs that provides insight into the well-being of the breeding colony under current husbandry practices and conditions. We also documented the nutritional status of EISs maintained on mixed whole-prey diets of rodents, birds, and frog legs and explored the potential of an experimental sausage of pork, rabbit, and alligator formulated to match the nutritional profile of the average snake consumed by free-range EISs. Not only are there no appreciable ill effects associated with the consumption of atypical prey in an atypical format, but current diet plans for EISs also appear to be a sufficient source of vitamin E and selenium. However, further research is still needed on the vitamin D_3_ status of snakes, given the large discrepancy between free-ranging and captive EISs, while a longitudinal study with juvenile snakes maintained on basal and experimental diets could further elucidate the effect of diet on body condition and reproductive fitness.

## Figures and Tables

**Figure 1 animals-14-03324-f001:**
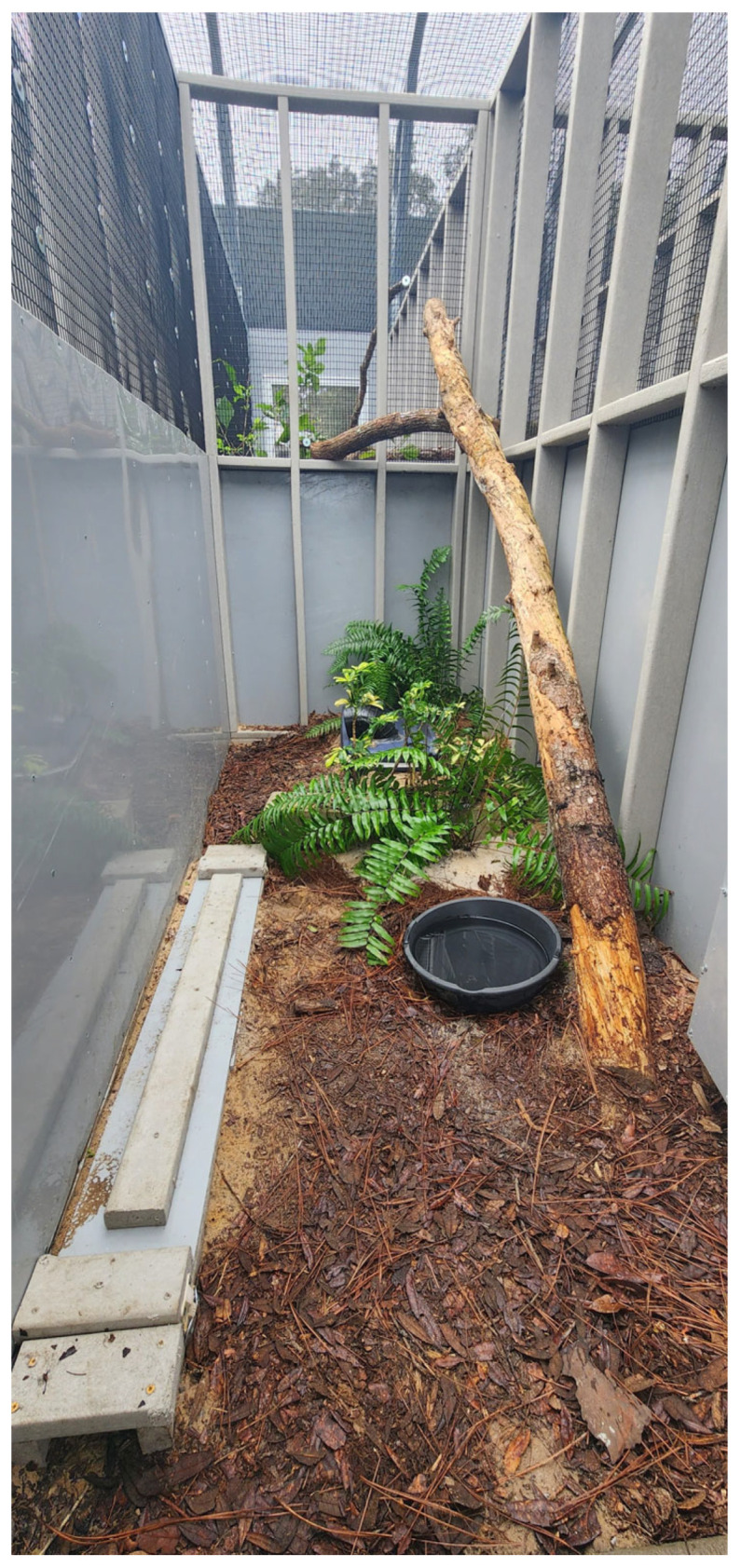
Outdoor enclosure for eastern indigo snakes (*Drymarchon couperi*) housed at the Orianne Center for Indigo Conservation, Eustis, FL, USA.

**Figure 2 animals-14-03324-f002:**
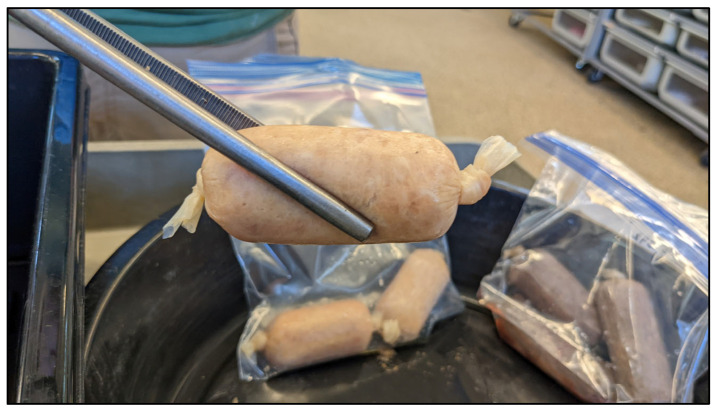
An approximately 60 g “faux-snake” sausage comprising pork loin, rabbit, and alligator filet in a 32 mm sausage casing.

**Figure 3 animals-14-03324-f003:**
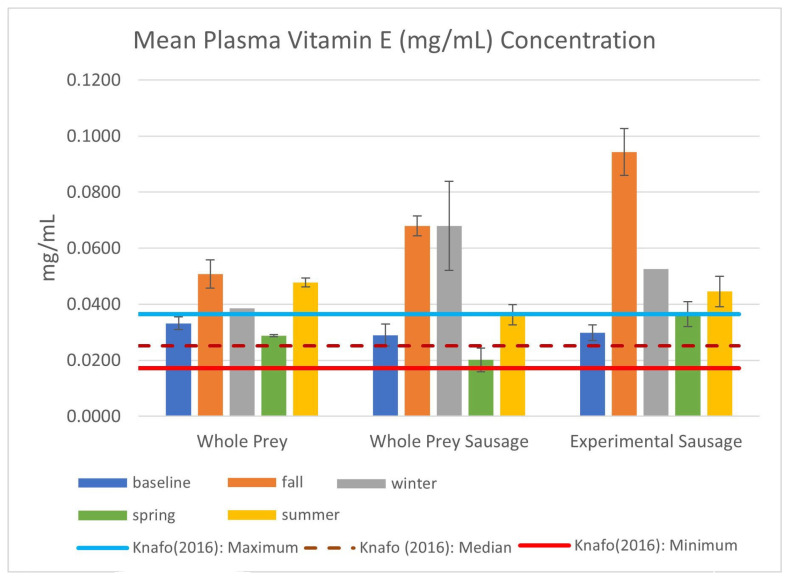
Mean vitamin E (-tocopherol; mg/mL) plasma concentration in EISs consuming whole prey or experimental diets between August 2022 and September 2023, including standard error bars and descriptive statistics from Knafo et al. (2016) [28].

**Figure 4 animals-14-03324-f004:**
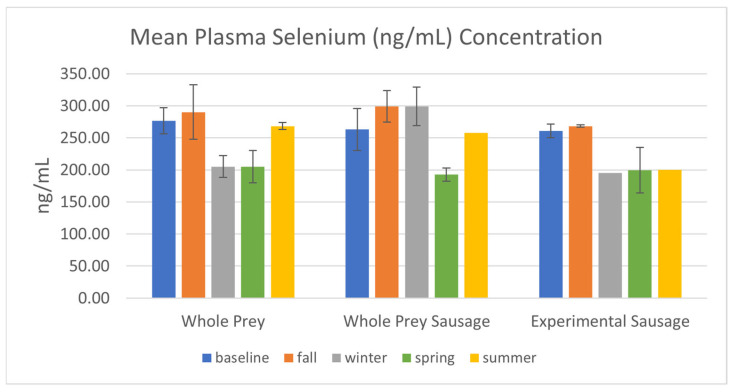
Mean selenium (ng/mL) plasma concentration in EISs consuming whole-prey or experimental diets between August 2022 and September 2023, including standard error bars.

**Figure 5 animals-14-03324-f005:**
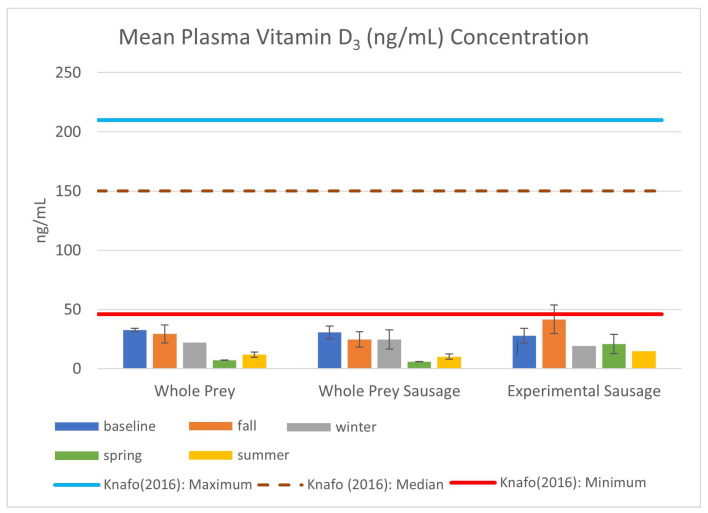
Mean vitamin D_3_ (ng/mL) plasma concentration in EISs consuming whole-prey or experimental diets between August 2022 and September 2023, including standard error bars and descriptive statistics from Knafo et al. (2016) [28].

**Table 1 animals-14-03324-t001:** Calculated nutritional profile of the experimental sausage diet composed of pork loin, rabbit tenderloin, and alligator filet supplemented with MeatComplete with Taurine, dicalcium phosphate, and uniodized salt.

Nutrient		Zootrition Calculations		Recipe A		Recipe B	
	Units	Dry Matter Basis	As Fed Basis	Dry Matter Basis	As Fed Basis	Dry Matter Basis	As Fed Basis
Crude Fat	%	8.85	2.93	6.95	2.06	11.70	3.30
Crude Fiber	%	0.054	0.018	NR	NR	NR	NR
Crude Protein	%	70.84	23.44	69.3	20.5	71.2	20.5
Taurine	%	0.33	0.108	NR	NR	NR	NR
Metabolizable Energy *	kcal/g	3.74	1.24	4.612	1.367	5.110	1.411
Ash	%	11.96	3.96	18.99	5.63	16.08	4.53
Calcium	%	4.489	1.46	4.40	1.30	4.01	1.13
Copper	mg/kg	27.24	9.02	32	10	17	5
Iron	mg/kg	44.52	14.73	126	37	80	23
Magnesium	%	0.12	0.040	0.10	0.03	0.10	0.03
Manganese	mg/kg	79.77	26.37	81	24	43	12
Phosphorus	%	2.23	0.76	2.63	0.78	3.27	0.92
Potassium	%	1.14	0.37	0.89	0.26	1.18	0.33
Selenium	mg/kg	NR	NR	0.53	0.15	0.87	0.25
Sodium	%	0.60	0.20	0.551	0.163	0.914	0.258
Zinc	mg/kg	50.10	16.58	44	13	53	15
Vitamin A **	IU/g	57.57	19.05	55.50	25.70	41.10	11.80
Vitamin E ***	mg/kg	401.34	134.82	926	262	453	130
Vitamin D_3_	IU/g	6.02	1.10	9.330	2.640	3.620	1.04

NR = Not reported; * DairyOne reported value as Gross Energy; ** retinol; *** α-tocopherol.

**Table 2 animals-14-03324-t002:** Nutritional composition of whole and partial prey items offered to carnivores in captivity, snakes observed in the stomach contents of free-range eastern indigo snakes (EISs), and the faux snake sausage presented in this case study.

Prey Item (g/serving)	Adult Mouse (20 g) [10]	Adult Rat (55 g) [10]	Day-Old Quail (10 g) [10]	Day-Old Chick (32 g) [10]	American Bullfrog Legs (70 g) [27]	Snakes Predated by EIS [29]	Experimental Sausage (65 g)
Gross Energy (kcal/g)	5.25	6.37	6.79	5.8	3.05	NR	5.11
% Protein	55.8	61.8	24.74	64.9	16.4	72.89	71.2
% Fat	23.6	32.6	11.04	22.4	0.3	7.28	11.7
Vitamin E (IU/kg)	100.4	139.2	NR	NR	10	306.02	453
Ca:P Ratio	1.38	1.8	NR	1.43	0.05	1.73	1.23

NR = Not reported.

## Data Availability

Raw data are available.

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
