# Peer review of "Evaluation of Nutritional and Health Status in Captive Eastern Indigo Snakes (Drymarchon couperi) in Response to Formulated Sausage Diet"

_animals, 2024, doi:10.3390/ani14223324_

Round 1

Reviewer 1 Report

Comments and Suggestions for Authors

The article entitled „Evaluation of Nutritional and Health Status in Captive Eastern Indigo Snakes (Drymarchon couperi) in Response to Formulated Sausage Diet“ is a case report of a pilot study focused on threatened eastern indigo snake. A collective of authors described the development of formulated diets and their effect on the nutritional status of examined individuals. The paper brings very interesting information for the care sheet of this species. Nevertheless, I feel it is necessary to incorporate some of my suggestions to improve its scientific soundness. According to my comments below, I recommend a major revision.

Please find my comments below.

Major comments:

·         In the abstract and throughout the manuscript, it is written that you studied the blood cell counts and 22 other biochemical items from plasma via biochemistry analysis. Nevertheless, the results show only values for vitamin E, D3 and selenium. Please add blood cell counts and other evaluated values or remove the information from the text.

·         You stressed under the abstract and introduction that females suffer from egg-binding. Is this issue somehow correlated with analysed values? This information will support the intention of the paper. You mentioned only briefly in the introduction and discussion that it can influence the reproductive status. It is a very general statement and the examples are given for mammals. You should focus more on squamates and specific problems of egg-binding. From the paper, it is not clear to me, which values should be monitored. Those data should be available in the veterinary literature I guess.

·         Were the snakes from nature or captive-bred?

·         What is the baseline in charts depicting the values of vitamin E, D3 and selenium? What are whiskers – 95% C.I., standard errors, min-max range? It must be written. You must also explain all abbreviations (e.g., WP). The reference Knafo should be Knafo et al. 2016 if you mean that from the reference list (no. 29).

·         Section 3.1: it seems that snakes were supplemented with vitamin E but there is nothing about it in the methodology. When and how often it was applied? What about the dosage? Did they also get other vitamins (e.g., vitamin D3)?

Minor comments:

·         Caption of Figure 1. The Latin name should be in italics.

·         Footnote of Table 1: α-tocopherol should have three asterisks.

·         Results, second row: should be figure 4, not 3.

·         Results, the last but one row of the first paragraph – study populations should be replaced by study groups

·         Results, Figure 5 should be 6; Figure 6 should be 7

·         The link to Figure 4 – the photo of the snake is not in the text

·         Several issues concerning the style of references should be managed. I found some mistakes. Authors should also focus on the journal reference style. The names of the journals should be abbreviated with dots – here it is rather random. Ref. 20 – the missing name of the journal; Latin names of animals – only the name of the genera should be capitalized; dois are missing (e.g. ref. no 37 - 10.7589/0090-3558-47.1.107).

Comments on the Quality of English Language

 I am not a native speaker but sometimes I have a problem with understanding the text. Some expressions are not standard for scientific texts (e.g. the use of dams instead of females). There is no line numbering so it is hard to write all the inaccuracies…I suggest revising the text again according to the typing errors, language and scientific terms.

Author Response

Comment 1:

In the abstract and throughout the manuscript, it is written that you studied the blood cell counts and 22 other biochemical items from plasma via biochemistry analysis. Nevertheless, the results show only values for vitamin E, D3 and selenium. Please add blood cell counts and other evaluated values or remove the information from the text.

Response 1:

Thank you for pointing that out. We decided to provide the blood cell counts and other evaluated values as supplemental information; the text was updated throughout to distinguish between results presented and the full suite of tests run (lines 238-243;258-264),

Comment 2:

You stressed under the abstract and introduction that females suffer from egg-binding. Is this issue somehow correlated with analysed values? This information will support the intention of the paper. You mentioned only briefly in the introduction and discussion that it can influence the reproductive status. It is a very general statement and the examples are given for mammals. You should focus more on squamates and specific problems of egg-binding. From the paper, it is not clear to me, which values should be monitored. Those data should be available in the veterinary literature I guess.

Response 2:

We provided additional information on the research available for reptile nutrition and reproductive health throughout the introduction and discussion. We also reorganized the introduction and discussion to improve the flow between ideas and better establish context for this study. 

The second paragraph of the introduction establishes context with how current husbandry practices may redispose snakes to obesity (line 56) and how this differs from those for other reptiles before diving into specific nutritional disorders such as hypovitaminosis E/metabolic bone disease. 

Response 3:

  • Were the snakes from nature or captive-bred?
  • What is the baseline in charts depicting the values of vitamin E, D3 and selenium? What are whiskers – 95% C.I., standard errors, min-max range? It must be written. You must also explain all abbreviations (e.g., WP). The reference Knafo should be Knafo et al. 2016 if you mean that from the reference list (no. 29).
  • Section 3.1: it seems that snakes were supplemented with vitamin E but there is nothing about it in the methodology. When and how often it was applied? What about the dosage? Did they also get other vitamins (e.g., vitamin D3)?

Response 4:

  • The snakes in this study were captive bred.
  • Figures were clarified to include explanations for abbreviations, the complete reference for ref 29, and whiskers (Standard Error). 
  • This was elaborated upon in lines 265-270. 

Minor comments // Response

  • Caption of Figure 1. The Latin name should be in italics. // Latin name italicized. 
  • Footnote of Table 1: α-tocopherol should have three asterisks. // Asterisk added. 
  • Results, second row: should be figure 4, not 3. // Figures relabeled following comment by another reviewer. 
  • Results, the last but one row of the first paragraph – study populations should be replaced by study groups // Changed. 
  • Results, Figure 5 should be 6; Figure 6 should be 7 // Figures relabeled following comment by another reviewer. 
  • The link to Figure 4 – the photo of the snake is not in the text // Photo removed following comment by another reviewer. 

Response 5: 

  • Several issues concerning the style of references should be managed. I found some mistakes. Authors should also focus on the journal reference style. The names of the journals should be abbreviated with dots – here it is rather random. Ref. 20 – the missing name of the journal; Latin names of animals – only the name of the genera should be capitalized; dois are missing (e.g. ref. no 37 - 10.7589/0090-3558-47.1.107).

Comment 5:

Revisited reference management software to manually update/change references in accordance with journal style. 

Reviewer 2 Report

Comments and Suggestions for Authors

Evaluating and improving nutrition and health in captive snakes is very important. Such knowledge can positively impact breeding programs of endangered species and the husbandry of any snake, whether kept by professionals in Zoos or by private people (citizen conservation).

The paper reminded me of similar approaches in the 1970s, when turtle keepers invented a gelatinous pudding for turtles with similar goals in mind.

The authors mention that egg-binding is a major problem in captive Indigo snakes, and their goal is to duplicate the nutritional profile of wild prey items consumed by EIS and to evaluate the effects of the sausage diet on overall body condition and a better understanding of linkages between diet and reproductive output in this species. However, even though six adult females were among the tested snakes, zero information on reproductive output was provided. I find this very disappointing. Here the authors should either include any information they have recorded (more egg-binding, less egg-binding?) or delete the goal of “reproductive output” from the introduction. If the authors plan to include reproduction, then general information about the reproductive behavior of the Indigo snake should be shortly provided in the introduction.

The recipe and preparation process for the sausage are well-described. Here the authors should include any history of previous published recipes for sausage productions for feeding snakes or reptiles. For example, the internet offers “Reptilinks” sausages for diverse reptiles in diverse nutritional assemblages. In Figure 3, the authors should provide measurements for length and diameter and the material for the sausage casing (pork intestine?). In my opinion, figures 2 and 3 can be combined into one figure. In Figures 5 and 7, the red horizontal line in the legend should be moved down next to “Mean” to be separated from the column color legends.

Author Response

Comment 1:
Evaluating and improving nutrition and health in captive snakes is very important. Such knowledge can positively impact breeding programs of endangered species and the husbandry of any snake, whether kept by professionals in Zoos or by private people (citizen conservation).

The paper reminded me of similar approaches in the 1970s, when turtle keepers invented a gelatinous pudding for turtles with similar goals in mind.

Response 2: 

Thank you for mentioning the turtle gel - it's still very much in use today and has expanded to include products for other animals like amphibians, fish, and even invertebrates. There are plenty of brands to chose from now, but some keepers have made species-specific recipes (ie blue death-feigning beetles come to mind). We added a brief mention of this in the discussion (starting line 316)

Comment 2:

The authors mention that egg-binding is a major problem in captive Indigo snakes, and their goal is to duplicate the nutritional profile of wild prey items consumed by EIS and to evaluate the effects of the sausage diet on overall body condition and a better understanding of linkages between diet and reproductive output in this species. However, even though six adult females were among the tested snakes, zero information on reproductive output was provided. I find this very disappointing. Here the authors should either include any information they have recorded (more egg-binding, less egg-binding?) or delete the goal of “reproductive output” from the introduction. If the authors plan to include reproduction, then general information about the reproductive behavior of the Indigo snake should be shortly provided in the introduction.

Response 2: 
Thank you for pointing out the disconnect in our introduction and overall study purpose - while our ultimate goal is to improve the reproductive output of EIS, we cannot fully realize the linkage without longitudinal studies initiated with juvenile subjects throughout sexual maturity (line37,306,402). Rather this pilot study was designed to evaluate the health and nutrition status of managed EIS with current practices (line 15,18) and explore the potential of formulated diet with atypical prey(line 21). In addition to palatability and practicality, we wanted to ensure that there were no ill effects associated with improper assimilation of nutrients due to any aspect of the sausage diet (meat-type, size/amount, etc). 

We clarified this by removing mentions of reproductive output and focusing on broader terms ie fitness, health (lines 112); as well as providing additional discussion on reptile nutrition (lines 28) and feeding management practices (lines 318, other alternative diets and studies with lines 323+). We elaborate on obesity/nutritional diseases in reptiles (lines 56,76 for piscivorous fish and a case of hypovitaminosis E), and elaborate on nutrient composition of domestic vs free-range whole prey (line 340).

Comment 3:

The recipe and preparation process for the sausage are well-described. Here the authors should include any history of previous published recipes for sausage productions for feeding snakes or reptiles. For example, the internet offers “Reptilinks” sausages for diverse reptiles in diverse nutritional assemblages.

Response 3:

Thank you - we did tighten up the language in the methodology and shift much of the content into a supplemental document based on feedback from another reviewer, but we also feel that this provided us with additional space to expand upon tips, troubleshooting, recipes, etc. We did expand upon other examples of sausage diets for snakes, but did not wish to touch on commercial products as we did not perform any traditional analyses. We did provide additional context for partial prey items given the increased prevalence of frog legs in reptile diets and available research on the nutrient composition of complete and incomplete whole anurans (line 327). 

Comment 4:

In Figure 3, the authors should provide measurements for length and diameter and the material for the sausage casing (pork intestine?). In my opinion, figures 2 and 3 can be combined into one figure. In Figures 5 and 7, the red horizontal line in the legend should be moved down next to “Mean” to be separated from the column color legends.

Response 4: 

Additional information (size,diameter,etc) were expanded upon in the methodology and supplemental file. 

Reviewer 3 Report

Comments and Suggestions for Authors

- Recommended to rearrange the keywords in alphabetical order.

- I suggest using "basal" instead of "baseline." Additionally, more details about the experimental treatments are required for better clarity.

- The quality of the pictures included in this manuscript is low. Please update them with higher-resolution images.

- Recommend moving the sausage pictures to the Supplementary Data section. The same applies to Figure 4. These images do not appear to contribute to the academic or scientific quality of the manuscript.

- The experimental design is not appropriate. The authors should clarify how many EIS were used in each group and whether there were any replicates. Additionally, why were snakes randomly assigned to one of three diet treatments? The choice of experimental animals is also questionable, as the group consists of three males and six females. How did the authors ensure that feed intake would be consistent across genders?

- Why did the authors only test for Vitamin E and selenium? Are these parameters sufficient to support the research hypothesis? Please provide clarification.

- The comparison made here is inconsistent. The feed intake mechanisms between elephants and snakes may differ significantly. Since the current study did not investigate this parameter, it is not appropriate to reference it in the context of your findings.

- The current study provides limited data, yet the authors discuss several findings from previous publications that are not directly relevant to their research. This should be reconsidered. The manuscript reads more like a protocol for producing sausage products rather than a scientific research article. The authors should review and revise the manuscript to ensure it fits the format and expectations of a research article.

Author Response

Comment 1:

- Recommended to rearrange the keywords in alphabetical order.

Response 1:

Agree, keywords reorganized.

Comment 2:

  • I suggest using "basal" instead of "baseline." Additionally, more details about the experimental treatments are required for better clarity.

Response 2:

Thank you for the suggestion - it was helpful in distinguishing between the basal sausage group and baseline sampling event. We provided additional details such as the demographics of each treatment group (123-128) and husbandry for the colony (130-145). 

Comment 3:

The quality of the pictures included in this manuscript is low. Please update them with higher-resolution images.  Recommend moving the sausage pictures to the Supplementary Data section. The same applies to Figure 4. These images do not appear to contribute to the academic or scientific quality of the manuscript.

Response 3:

Photographs were re-uploaded and higher-quality images provided. Photographs and detailed methodology were incorporated into the supplemental text. 

Comment 4:

The experimental design is not appropriate. The authors should clarify how many EIS were used in each group and whether there were any replicates. Additionally, why were snakes randomly assigned to one of three diet treatments? The choice of experimental animals is also questionable, as the group consists of three males and six females. How did the authors ensure that feed intake would be consistent across genders?

Response 4:

Additional information was provided regarding the experimental design and decisions re: diet treatments (lines 123-127). The original experimental design included far more animals but due to budget restraints the sample size was cut in half and female animals prioritized given the focus on reproductive health. While the snakes were bred in-house at the OCIC, there are still variations in their husbandry that cannot be controlled for (age, reproductive history, health conditions, husbandry and management) that create an additional layer of uncertainty when interpreting data collected from a small group. 

We clarified the distinction between our ultimate goal (improved reproductive health and nutrition for managed EIS) with the pilot study's objective (how do EIS fare under current husbandry practices and are atypical diets safe? - lines 19,21,33,36,112-119,305-311,364) These animals are also active participants in the OCIC's breeding colony and husbandry decisions were largely unchanged (line 130-145): animals were fed according to individual needs/seasonal indicators. Individuals may have different preferences for thermal gradients for digestion or different responses to reproductive cues, leading to differences in feed intake. Snakes fed, bred, and behaved otherwise normally on both basal and experimental sausages.  

Comment 5:

Why did the authors only test for Vitamin E and selenium? Are these parameters sufficient to support the research hypothesis? Please provide clarification. The comparison made here is inconsistent. The feed intake mechanisms between elephants and snakes may differ significantly. Since the current study did not investigate this parameter, it is not appropriate to reference it in the context of your findings.

Response 5:

Vitamin D3 was also tested in this study section 3.1; line 282). These parameters were selected given previous research in animal nutrition regarding these nutrients and reproductive health (lines 66-75, 83). There is relatively little information regarding snake nutrition, but by using what we know about model carnivores and nutrient supplements in other species we can begin to understand where potential shortcomings in husbandry may be occurring and explore potential methods to address them - such as formulated diets. 

Comment 6:

The current study provides limited data, yet the authors discuss several findings from previous publications that are not directly relevant to their research. This should be reconsidered. The manuscript reads more like a protocol for producing sausage products rather than a scientific research article. The authors should review and revise the manuscript to ensure it fits the format and expectations of a research article.

Response 6:

Thank you for this valuable feedback - upon further review, we do agree that too much of the focus was on making sausages and that other areas could be further expanded upon to establish the context for this research. We also clarified that our results are not only a pilot study, but also that the focus is on the health/nutrition status of EIS under current management practices. Much of this was addressed in previous responses/highlighted in the marked manuscript. 

We are lucky enough to have numerous studies in which the nutrient compositions of free-range and captive prey items are analyzed, including one specific to EIS, and a data set on the health/nutrition status of free-range EIS. The findings of these studies are foundational to the methodology of this project, and other research into mammalian/carnivore nutrition can provide guidance on potential areas of concern when used in conjunction with free-range EIS health/nutrition statuses.

Round 2

Reviewer 1 Report

Comments and Suggestions for Authors

I was happy to read a new version of the manuscript. Now it is clear and easy to follow. I have just two minor comments:

Line 282, 301: descriptive (not description) statistics

Line 266: Are still important FOR understanding

Reviewer 3 Report

Comments and Suggestions for Authors

Dear Authors,

Thank you for your responses. The manuscript has been improved.

All the comments and suggestions have addresses carefully. I totally agree with their responses.

There is no more additional issues from me.

Thank you and best wishes.